# Visible-Light-Promoted Tandem Skeletal Rearrangement/Dearomatization of Heteroaryl Enallenes

**DOI:** 10.3390/molecules29030595

**Published:** 2024-01-25

**Authors:** Maurizio Chiminelli, Gabriele Scarica, Andrea Serafino, Luciano Marchiò, Rosanna Viscardi, Giovanni Maestri

**Affiliations:** 1Dipartimento di Scienze Chimiche, della Vita e della Sostenibilità Ambientale, Università di Parma, Parco Area delle Scienze 17°, 43124 Parma, Italy; maurizio.chiminelli@unipr.it (M.C.); gabriele.scarica@unipr.it (G.S.); andrea.serafino@unipr.it (A.S.); luciano.marchio@unipr.it (L.M.); 2ENEA, Casaccia Research Center, Santa Maria di Galeria, 00123 Roma, Italy; rosanna.viscardi@enea.it

**Keywords:** allenes, visible light, energy transfer, cycloadditions, dearomatization

## Abstract

Access to complex three-dimensional molecular architectures via dearomatization of ubiquitous aryl rings is a powerful synthetic tool, which faces, however, an inherent challenge to overcome energetic costs due to the loss of aromatic stabilization energy. Photochemical methods that allow one to populate high-energy states can thus be an ideal strategy to accomplish otherwise prohibitive reaction pathways. We present an original dearomative rearrangement of heteroaryl acryloylallenamides that leads to complex fused tricycles. The visible-light-promoted method occurs under mild conditions and tolerates a variety of functional groups. According to DFT modeling used to rationalize the outcome of the cascade, the reaction involves a sequential [2+2] allene–alkene photocycloaddition, which is followed by a selective retro- [2+2] step that paves the way for the dearomatization of the heteroaryl partner. This scenario is original with respect to the reported photochemical reactivity of similar substrates and thus holds promise for ample future developments.

## 1. Introduction

Access to complex molecular architectures from simple, cheap and readily available building blocks is a recurrent focus of modern synthetic chemistry. Ever since the publication of the popular article by Lovering [1] on the relevance of saturated carbon atoms in medicinal chemistry, the scientific community has devoted increasing interest to the development of synthetic methods selectively transforming aryl and alkenyl C(sp^2^) nuclei into corresponding C(sp^3^) ones. This strategy could possibly increase the success rate of potential drug candidates [2], and it is at the core of several recent research projects [3,4].

In this context, methods that afford fused/bridged polycycles through the dearomatization of ubiquitous aryl rings are currently in high demand [5,6,7,8]. These strategies invariably face challenges connected with the loss of the aromatic stabilization of the arene starting material. In order to limit the energetic burden connected with this step, in recent years, several approaches based on the exploitation of visible light have emerged as a promising tool to expand the boundaries of dearomative cycloadditions [9,10]. The use of photochemical approaches could indeed lead to the formation in solution of high-energy species, which could, thus, reduce the energetic cost connected with the loss of aromatic stabilization.

Visible-light-promoted dearomatization methods involve either the activation of the arene partner [11,12,13] or the activation of the arenophile one [14,15,16], and the latter approaches are less explored. We present herein a complementary approach, in which the key dearomative step occurs downstream from a [2+2] photochemical cycloaddition. The ordered cascade could thus lead to fused hetero-tricyclic frameworks that would be otherwise difficult to prepare.

The reaction involves the presence of three different functional groups in the starting material, namely an allenamide unit, an acryloyl fragment, and a five-membered O- or S- containing heteroaromatic ring. The alkene and the allene partners are common functional groups for photochemical [2+2] cycloadditions (Figure 1, way a) [17,18,19]. In these strategies, the fine tuning of the structure of the polyunsaturated reagent is crucial to trigger a reaction in a chemo- and regio-controlled fashion. For instance, we reported that 1,7-enallenes could selectively afford the corresponding [3.2.0] bicycles with a pendant vinylidene arm [20], while, more recently, the preparation of [4.2.0] bicycles with an endocyclic double bond had been disclosed [21].

The presence of significant strain energy in rigid bicyclic scaffolds that can form upon photochemical cycloadditions could represent a useful tool to unleash additional reaction pathways. In the context of [2+2] cycloadditions, a key example to this end has been reported in the cascade of enynes reported by Park (Figure 1, way b) [22]. According to his experimental and computational studies, the reaction of 1,7-enynes leads to the intermediate formation of a strained bicyclic compound, which then undergoes a selective retro- [2+2] that eventually affords the desired product a 1,3-conjugated diene unit.

As part of an ongoing interest in the development of photochemical cascades that proceed with complete atom economy [24,25,26], we recently disclosed a method for the formal dimerization of enallenes that affords complex molecular architectures through the controlled rearrangement of 12 π-type electrons (Figure 1, way c) [23]. The main product of these reactions had a [3.2.0] bicyclic unit tethered to a fused tricycle, and the minor side-product of the sequence was the corresponding tricycle instead.

We present herein a method for the synthesis of fused tricyclic scaffolds through a visible-light-promoted cascade on heteroaryl-substituted enallene derivatives (Figure 1d). The reaction occurs with good efficiency and complete diastereocontrol under mild and practical conditions (40 °C, 3 h), affording complex molecular architectures that would otherwise be synthetically challenging. The method involves a skeletal rearrangement of the carbon nuclei involved in the π groups of the substrate and allows one to trigger the selective dearomatization of the five-membered heterocyclic ring. The sequence results in products featuring an alternate 1,4,7-triene motif. This is complementary to the skeletal rearrangement observed for the photochemical cascade of enyne derivatives that afforded a conjugated 1,3-arrangement of the C–C double bonds instead [22]. Moreover, the present cascade approach is an alternative to most dearomatization methods that exploit visible light because it does not involve the direct activation of a reacting partner to trigger a dearomative step [11,12,13,14,15,16]. Our approach shows instead the feasibility of devising a sequence that forms a stable, closed-shell intermediate, which is, in turn, prone to undergo a tandem dearomatization step driven by the release of its steric strain.

## 2. Results

In a typical experiment (Table 1, entry 1), 0.2 mmol of substrate **1a** (0.1 M in DCM) and 1 mol% of an Iridium-based photosensitizer were put in a standard 5-mm NMR tube, in order to maximize the surface/volume ratio of the solution. The tube was immerged in a silicon–oil bath thermostated at 40 °C and surrounded circa 10 cm by a 14 W blue LED strip (300 LEDs, photo in Appendix A), which has its maximum emission band centered at 455 nm [26]. The solution was irradiated for three hours, and the complete consumption of the starting material was monitored by TLC. The reaction led to the formation of the desired tricycle **2a** in 66% yield according to ^1^H NMR using an internal standard as reference. From the crude NMR spectrum, no significant additional resonances were observed, suggesting that mass balance could be accounted for by partial substrate decomposition, likely by oligomerization/polymerization as observed in related cascades. The reaction did not require the use of degassed solvents.

Several optimization tests were performed, and a summary of the main trend observed is provided in Table 1, while the complete list of performed essays is provided in the Appendix A.

The adopted dilution of **1a** proved optimal, and lower yields were measured either using a more diluted solution, which might erode the yield because a lower rate of the photocatalytic process would increase the impact of the spontaneous substrate decomposition, or by increasing the concentration (entry 2), which might favor the formation of dimeric/oligomeric species as previously observed using related aryl enallenes (Figure 1). An experiment performed at room temperature provided **2a** in a slightly diminished yield (entry 3). Various solvents were tested (entries 4–7), and no clear correlation between their polarity and the yield of the desired product emerged. The choice of the photocatalyst proved crucial to steer the selectivity of the reaction (entries 8–10). In general, various Ir(III) photosensitizers proved competent for the desired transformation, while other popular organic and organometallic derivatives yielded worst results. Among iridium complexes, the best result was achieved with the fluorinated derivative used in entry 1, which has the highest triplet energy of the series [27]. Finally, no reaction occurred without either the sensitizer or light (entry 11) or by performing the reaction at 10 °C. Once good conditions were secured, we then planned to study the scope of the reaction by preparing a small library of substrates featuring the three functional groups of the cascade (Figure 1).

In general, we planned to prepare our starting materials through a convergent approach.

Starting from heteroarenes that present an aldehyde group at their 2 position, which are either commercially available or can easily be accessed through a Vilsmeier reaction, we envisioned to establish the acryloyl unit through a Wittig reaction by treatment with a bromoacetate derivative. The transformation of the ester into an acyl chloride would then allow one to perform the reaction with a secondary amine featuring an *N*-propargyl arm, and the corresponding product would eventually be converted into the desired starting material by means of a base-promoted alkyne-to-allene isomerization. Alternatively, from 2-halo heteroarenes, the intermediate acyl chloride derivative could be accessed using a Heck-type cross-coupling reaction using an acrylate. This synthetic plan led to the preparation of circa 20 enallenes **1**, which were then tested under the previously optimized conditions (Figure 2).

The reaction on the model substrate **1a** led to the isolation of **2a** in a 63% yield. The structure of the latter was unambiguously assigned thanks to single-crystal X-ray diffraction analysis, which confirmed the *anti*-arrangement between the C-H groups of the two contiguous stereocenters of the product. The model reaction could be scaled up to 1.4 mmol, without significant erosion of the yield. The use of a 15 mL sealed tube, which has a lower surface/volume ratio compared to an NMR tube, required, however, the irradiation of the reacting solution for a longer period (9 h) [24,28]. We next tested the effect of substituents of the benzyl group on the nitrogen atom. Gratifyingly, both electron-donating and electron-withdrawing groups were tolerated by the method (**2b**,**c**), which was thus suitable to treat acetal-protected diols and aryl bromides, which can be useful synthetic tools for further derivatizations. A slightly diminished yield was observed using an *N*-methyl substrate (**2d**). The replacement of the thienyl ring with a furyl one did not alter the outcome of the sequence (**2e**). The deuterium labeling of the C(sp^2^)-H group α- to the carbonyl unit led to the recovery of the corresponding product **2f**, in which the deuterium group was retrieved exclusively on the carbon atom bound to the carbonyl motif. Additional attempts to label the carbon atoms of the allenyl unit with *d*-nuclei were, however, fruitless at present. Furane-based substrates can tolerate various functionalities on the benzyl protecting group, such as fluorides and ethers (**2g**,**h**). Interestingly, an *N*-cyclopropyl derivative could be prepared in a synthetically useful yield (**2i**). Benzothiophene groups can be reacted, leading to the formation of the corresponding fused tetracyclic derivatives with good efficiency (**2j**). The presence of *meta*-substituents on the *N*-protecting group had a negative effect on the yield in this case (**2k**), possibly because of a negative steric clash with the heteroarene. The benzofurane scaffold was similarly tolerated (**2l**). The installation on the model substrate of an additional substituent on the C3 position of the heterocycle led to the recovery of the corresponding product that featured a challenging quaternary carbon at one headbridging position. Remarkably, the product **2m** was recovered with the highest yield of the series, suggesting that the bis-allylic nature of the corresponding tertiary C-H group of products **2** might have been the cause of a partial yield loss because of their reduced bond dissociation energy [29]. Finally, we prepared enallene **1n**, in which the heterocycle was bound to the alkene through its C3 position rather than the C2 one. However, no traces of the corresponding product were observed, and extensive substrate decomposition occurred in this case.

The preparation of substrates **1**, in which a second electron-withdrawing group such as a BOC or a tosyl one was bound to the nitrogen atom of the amide, was limited by the reduced tendency of the corresponding alkyne precursors to undergo final isomerization. Upon several failed attempts, a substrate in which the benzyl group of **1a** was replaced by a BOC one could be isolated. However, under the optimized photocatalytic conditions (Table 1), no traces of **2** were observed, and the substrate underwent extensive decomposition. These results suggest that the present method is not suitable to react enallenes in which the allenamide arm is substituted by two electron-withdrawing groups.

We then attempted the dearomatization of nitrogen-containing heterocycles. The reaction of pyrrole-derived substrate **1o** led to the formation of tricycle **2o**, which was established in a clear-cut fashion via X-ray diffraction analysis. This result showed that the cascade observed for *S*- and *O*-heterocycles is not accessible with *N*-derivatives. Because these classes of derivatives have similar aromatic stabilization energies [2,23,30], we assumed that the divergent behavior of *N*-heterocycles might be due to steric factors stemming from the presence of the substituent on the heteroatom. The outcome observed with the pyrrole-based reagent was confirmed by reacting the corresponding indole derivative. The fused tetracycle **2p** was recovered in good yield, confirming that the reactivity of 5-membered *N*-heteroaryls bound to the alkene partner preferentially yielded a product that was similar to those previously observed in related dimerization cascades (Figure 1, way c).

In order to rationalize the mechanism of the cascade, the reaction was studied by DFT (Figure 3).

Calculations were performed at the M06/def2-TZVP level [31,32], which proved a reliable method to assess the pathways of related photochemical cascades [15,23,24], using dichloromethane as an implicit solvent [33]. According to the literature, both the allenamide and the cinnamamide groups of enallenes **1** are redox-neutral in the range of potentials of common photocatalysts, including the Ir(III) complexes used in this present study [20,21,22,23,26,27]. Substrate **1** can be activated by an excited iridium photosensitizer via energy transfer (eT) [34,35,36], analogous to similar reactions involving an acylallenamide fragment [20,23,26]. The triplet energy of the corresponding intermediate **I** is +41.2 kcal/mol in ΔG, which is a value well within those accessible by the Ir complex [27]. The lowest energy triplet has its spin density localized on the former allenyl arm, as witnessed by the calculated spin density (lower part of Figure 3), which shows the perpendicular arrangement of the two mono-occupied molecular orbitals of the intermediate. The triplet can evolve via *5-exo-trig* cyclization. This step occurs through low-barrier **TS** (**I-II**) and affords intermediate **II**, in which the two radicals are in an allylic and benzylic resonance form, respectively. The stabilization of these two mono-occupied orbitals is responsible for the relatively low energy of the triplet, making this step largely exergonic. Spin-relaxation by intersystem crossing (ISC) gives closed-shell intermediate **III**, which lies slightly below the entry channel (−4.9 kcal/mol in ΔG).

In agreement with our previous studies on related aryl-enallenes (Figure 1), the localization of the two spins on the former alkenyl carbon atoms would provide triplet **I’**, which is less stable than **I** (by +4.4 kcal/mol in ΔG, data in the Appendix A). This suggests that the initial activation of the allenyl arm is more favorable. It is worth noting that, if the energy transfer step could cause the population of triplet **I’**, this species would still evolve into intermediate **II** via radical *5-exo-dig* cyclization, eventually following the proposed pathway for the skeletal rearrangement cascade.

The [3.2.0] bicyclic core of **III** is characterized by a significant steric strain, which is due to the presence of the endocyclic double bond that is bound to the cyclobutane ring. This strain explains the relatively high energy of **III** compared to **1**, considering that two π-bonds of the latter become σ-ones in the former. Indeed, similar alkene–allene [2+2] photocycloadditions, in which products are less congested, are invariably much more exergonic (Figure 1, way a).

The intermediate **III** could then evolve through two different manifolds. The enamine group can be activated by eT in the presence of a photoexcited Ir species. Similar reactivities have been reported in the literature [37]. The triplet energy of **^3^III** is +49.1 kcal/mol in ΔG. The value is higher than that observed for substrate **1** but still within reach of the iridium photosensitizer [27]. The cyclobutane ring can then undergo ring-opening because of the presence of a radical at its α- position. The process occurs through the low-barrier **TS** (**III,IV**) and affords intermediate **IV**, in which the two mono-occupied molecular orbitals are stabilized by an allylic- and a benzylic-like conjugation, respectively. The former could then attack the aryl ring via *6-endo/exo-trig* cyclization, delivering endergonic triplet **V**. This step occurs through **TS** (**IV,V**) (ΔG = +32.5 kcal/mol), which represents a relatively high barrier for a process that occurs at 40 °C. In intermediate **V**, the two mono-occupied molecular orbitals are arranged in a nearly perpendicular fashion in order to minimize spin repulsion. This species could then afford the desired product **2** via ISC. Because this pathway would involve a relatively high-energy transition state, we tried to model more feasible alternatives. The closed shell intermediate **III** might undergo a C–C cleavage, affording the biradical singlet **^1^IV**. The process occurs via **^1^TS** (**III,IV**), which has a ΔG of +16.2 kcal/mol. Although this step is more energy demanding than the corresponding one that occurs at the triplet state, the barrier is still remarkably low for the cleavage of a σ- C–C bond. It is worth noting that the geometry of **^1^IV** and **^3^IV** are nearly superposable. Moreover, the localization of their two mono-occupied molecular orbitals is nearly identical, too. As a result, it can also be conceived that the formation of **^1^IV** might occur via ISC from **^3^IV**. The former biradical singlet intermediate could then directly afford the desired product **2** through **TS** (**IV-2**). This step has a relatively low barrier for dearomative cyclization (ΔG = +10.9 kcal/mol), which makes it much more energetically convenient than the above-mentioned triplet alternative. By analyzing the population of the optimized **TS** (**IV-2**), it can be observed that the singlet biradical character is essentially lost in the TS. Indeed, the two HOMOs of its α and β electrons are superposable, and, together, they represent the electronic reorganization typical of a radical recombination process, which eventually affords the final product **2**. This is further shown by the partial orbital overlap in the TS between the β-carbon of the lactam ring and the C3 of the heterocycle (green basin in the corresponding picture of Figure 3).

The final product features a carbon skeleton that underwent reorganization compared to the starting material because of the ring-opening of an intermediate cyclobutane ring, which was favored by the release of its steric strain. The process is conceptually similar to that observed in related photochemical cascades of 1,7 enynes (Figure 1, way b), but it leads to a skipped 1,4,7-triene motif in the present case, thanks to the tandem dearomatization of the heteroaryl partner.

Attempts to locate a concerted transition state for the conversion of **III** into **2** via 3,3-sigmatropic rearrangement were fruitless, likely because of the rigidity of **III** that prevents the required frontier orbital overlap for this concerted process.

The reaction of substrates featuring an *N*-heterocycle (**1o**, **1p**) led to the formation of a product that did not undergo any skeletal rearrangement (Figure 2). The structure of these tricyclic compounds was identical to that of the minor byproduct that was observed in our previous study (Figure 1). This suggests that the formation of **2o** and **2p** could result from the same mechanism described for related aryl-enallenes [23]. Alternatively, it can be conceived that intermediate **^3^II** might undergo a radical *6*-*endo*/*exo-trig* cyclization on the *N*-heterocycle, and the subsequent rearomatization would form **2o** and **2p**. Finally, we cannot exclude that a similar scenario might occur upon the formation of intermediate **III**, which could undergo a strain-release-driven C–C cleavage, reforming either **^3^II** or the corresponding biradical singlet. In both cases, the allyl radical arm would then attack the heterocyclic ring and eventually afford the final product via rearomatization.

## 3. Discussion

The cascades reported herein afford complex molecular architectures in a concise fashion and with complete atom economy. These structures can be of interest because they present a 1,4,7-triene motif, which might be further used to increase their molecular complexity [38]. The rationalization of the skeletal rearrangement observed in the present reactions showed that the dearomatization of the heteroaryl ring is triggered by the ring-opening of an intermediate fused bicycle, which is driven by the release of its steric congestion [39,40]. A similar manifold holds promise for vast opportunities in future development because it represents an additional tool for dearomative processes [5,6,7,8,9,10,11,12,13,14,15,16]. Indeed, while light-promoted strategies are based on the direct activation of an arenophile (alkene, azo, or allene group) or of a sufficiently conjugated arene (acyl-naphthalenes, quinolines), the present approach relies on the use of light to build up steric strain as a form of chemical energy into an intermediate species, eventually using its release to unleash the final functionalization of an arene partner. It is worth noting that, at present, a similar approach is, however, limited to the dearomatization of derivatives with a relatively limited aromatic stabilization [30]. Therefore, further development will likely aim to overcome these limitations and extend the concept to more challenging arenes.

## 4. Materials and Methods

All chemicals were purchased from commercial sources and used as received. Solvents were dried via passing through alumina columns using an Inert^®^ system and were stored under nitrogen. Chromatographic purifications were performed under gradient or an isocratic regime using mesh 60 silica gel. ^1^H NMR and ^13^C NMR spectra were recorded at 300 K on a Bruker 400 MHz spectrometer using the solvent as internal standard (7.26 ppm for ^1^H NMR and 77.00 ppm for ^13^CNMR for CDCl_3_). Reported assignments were based on decoupling, COSY, NOESY, HSQC, and HMBC correlation experiments. The terms m, s, d, t, q, and quint represent multiplet, singlet, doublet, triplet, quadruplet, and quintuplet, respectively, and the term br means a broad signal. MS analyses were recorded on an Agilent Mass Spectrometer, and exact masses were recorded on an LTQ ORBITRAP XL Thermo Mass Spectrometer (electrospray sources).

Calculations were performed using the Gaussian16 package [41], using the model described in the Results section. The geometry of all the intermediates and transition states was optimized without any constraint. The approximate starting geometry of TSs was located through relaxed scans of the corresponding putative reaction coordinate. Intermediates were characterized by the absence of imaginary frequencies in their Hessian matrix. TSs were characterized by the presence of a single imaginary frequency in their Hessian matrix, which corresponded to the vibration connecting the reagent to the product. Biradical singlets were modeled through broken symmetry formalism in combination with the use of an unrestricted DFT functional. The actual biradical character of the resulting species was then assessed by population analyses.

### 4.1. General Procedure for the Alkyne Isomerization Leading to Allenes ***1** (**GP-1**)*

The desired enyne (1 equiv.) and THF (0.20 M) were sequentially added to a Schlenk tube equipped with a magnetic stirring bar. ^t^BuOK (0.2 equiv.) was added and the resulting mixture was stirred at room temperature for 10 min. After complete conversion as monitored by TLC, 5 mL of a saturated NH_4_Cl solution was added. The mixture was extracted with EtOAc (31 × 5 mL); the organic layers were separated and dried over Na_2_SO_4_. The solution was concentrated under reduced pressure, and the crude product was purified by chromatography on silica gel (*n*-hexane/EtOAc gradient) to afford the corresponding enallene.

#### 4.1.1. (*E*)-*N*-Benzyl-*N*-(propa-1,2-dien-1-yl)-3-(thiophen-2-yl)acrylamide

Enallene **1a** was prepared following the general procedure GP-1 from the corresponding enyne (436.3 mg, 1.55 mmol). Yellow solid (292.7 mg, 67% yield). Two rotamers were observed due to the dynamic rotation of the amide. ^1^H NMR (400 MHz, CDCl_3_) δ 7.91 (d, *J* = 15.0 Hz, 1H RotA, 1H RotB), 7.81 (t, *J* = 6.5 Hz, 1H RotA), 7.41–7.20 (m, 7H RotA, 7H RotB), 7.10–7.00 (m, 1H RotA, 1H RotB), 6.94–6.78 (m, 2H RotB), 6.62 (d, *J* = 14.9 Hz, 1H RotA), 5.36 (d, *J* = 6.3 Hz, 2H RotA, 2H RotB), 4.88–4.78 (m, 2H RotA, 2H RotB). ^13^C NMR (101 MHz, CDCl_3_) δ 202.7, 202.6, 164.8, 164.7, 140.2, 137.6, 137.2, 137.1, 136.6, 130.8, 128.9, 128.4, 128.1, 128.0, 127.9, 127.5, 127.1, 126.2, 115.7, 115.6, 100.1, 87.7, 86.8, 49.3, 48.3. ESI-MS calculated for C_17_H_16_NOS [M + H]^+^; 282.09 found 282.41.

#### 4.1.2. (*E*)-*N*-(2-Bromobenzyl)-*N*-(propa-1,2-dien-1-yl)-3-(thiophen-2-yl)acrylamide

Enallene **1b** was prepared following the general procedure GP-1 from the corresponding enyne (432.3 mg, 1.2 mmol). Yellow solid (309.8 mg, 72% yield). Two rotamers were observed due to the dynamic rotation of the amide. ^1^H NMR (400 MHz, CDCl_3_) δ 7.98–7.87 (m, 1H RotA, 1H RotB), 7.83 (t, *J* = 6.4 Hz, 1H RotA), 7.65–7.52 (m, 1H RotA, 1H RotB), 7.45–6.85 (m, 6H RotA, 8H RotB), 6.44 (d, *J* = 15.0 Hz, 1H 1H RotA), 5.38–5.22 (m, 2H RotA, 2H RotB), 4.96–4.80 (m, 2H RotA, 2H RotB). ^13^C NMR (101 MHz, CDCl_3_) δ 202.2, 165.0, 164.7, 140.2, 140.1, 137.6, 137.0, 135.9, 132.8, 132.7, 131.0, 130.9, 129.0, 128.4, 128.2, 128.1, 128.0, 127.4, 127.2, 121.9, 115.1, 100.0, 87.9, 86.8, 49.5, 48.9. ESI-MS calculated for C_17_H_15_BrNOS [M + H]^+^; 360.01 found 360.46.

#### 4.1.3. (*E*)-*N*-(Benzo[d][1,3]dioxol-5-ylmethyl)-*N*-(propa-1,2-dien-1-yl)-3-(thiophen-2-yl)acrylamide

Enallene **1c** was prepared following the general procedure GP-1 from the corresponding enyne (292.8 mg, 0.9 mmol). Orange viscous oil (188.5 mg, 64% yield). Two rotamers were observed due to the dynamic rotation of the amide. ^1^H NMR (400 MHz, CDCl_3_) δ 7.87 (d, *J* = 15.0 Hz, 1H RotA, 1H RotB), 7.79–7.71 (m, 1H RotA), 7.37–7.17 (m, 2H RotA, 2H RotB), 7.07–6.98 (m, 1H RotA, 1H RotB), 6.87–6.55 (m, 4H RotA, 5H RotB), 5.98–5.87 (m, 2H RotA, 2H RotB), 5.37 (d, *J* = 6.3 Hz, 2H RotA, 2H RotB), 4.74–4.64 (m, 2H RotA, 2H RotB). ^13^C NMR (101 MHz, CDCl_3_) δ 202.6, 202.5, 164.7, 164.6, 148.2, 147.7, 147.0, 146.7, 140.2, 137.1, 136.6, 131.4, 131.1, 130.8, 128.1, 127.9, 121.6, 119.5, 115.7, 115.6, 108.8, 108.5, 108.0, 106.8, 101.2, 100.9, 100.0, 87.8, 86.8, 49.0, 48.0. ESI-MS calculated for C_18_H_16_NO_3_S [M + H]^+^; 326.08 found 326.56.

#### 4.1.4. (*E*)-*N*-Methyl-*N*-(propa-1,2-dien-1-yl)-3-(thiophen-2-yl)acrylamide

Enallene **1d** was prepared following the general procedure GP-1 from the corresponding enyne (153.9 mg, 0.75 mmol). Orange viscous oil (87.1 mg, 57% yield). Two rotamers were observed due to the dynamic rotation of the amide. ^1^H NMR (400 MHz, CDCl_3_) δ 7.93–7.78 (m, 1H RotA, 1H RotB), 7.73–7.64 (m, 1H RotA), 7.38–7.20 (m, 2H RotA, 2H RotB), 7.09–6.93 (m, 1H RotA, 2H RotB), 6.79–6.65 (m, 1H RotA, 1H RotB), 5.49–5.36 (m, 2H RotA, 2H RotB), 3.25–3.02 (m, 3H RotA, 3H RotB). ^13^C NMR (101 MHz, CDCl_3_) δ 202.8, 201.7, 164.8, 164.3, 140.3, 136.8, 136.1, 130.9, 130.7, 128.1, 127.8, 127.7, 115.4, 101.3, 100.3, 87.3, 86.7, 33.0, 31.8. ESI-MS calculated for C_11_H_12_NOS [M + H]^+^; 206.06 found 206.13.

#### 4.1.5. (*E*)-*N*-Benzyl-3-(furan-2-ySl)-*N*-(propa-1,2-dien-1-yl)acrylamide

Enallene **1e** was prepared following the general procedure GP-1 from the corresponding enyne (185.6 mg, 0.7 mmol). Orange solid (140.9 mg, 76% yield). Two rotamers were observed due to the dynamic rotation of the amide. ^1^H NMR (400 MHz, CDCl_3_) δ 7.80 (t, *J* = 6.5 Hz, 1H RotA), 7.63–7.19 (m, 7H RotA, 7H RotB), 6.99–6.89 (m, 2H RotB), 6.73 (d, *J* = 15.0 Hz, 1H RotA), 6.64–6.56 (m, 1H RotA, 1H RotB), 6.53–6.42 (m, 1H RotA, 1H RotB), 5.39–5.29 (m, 2H RotA, 2H RotB), 4.90–4.79 (m, 2H RotA, 2H RotB). ^13^C NMR (101 MHz, CDCl_3_) δ 202.8, 202.4, 164.9, 151.5, 144.3, 137.6, 137.1, 131.1, 130.5, 128.8, 128.7, 128.4, 128.1, 128.0, 127.4, 127.1, 126.2, 114.8, 114.6, 114.3, 112.3, 100.2, 99.9, 87.6, 86.8, 49.2, 48.2. ESI-MS calculated for C_17_H_16_NO_2_ [M + H]^+^; 266.11 found 265.72.

#### 4.1.6. (*E*)-*N*-Benzyl-3-(furan-2-yl)-*N*-(propa-1,2-dien-1-yl)acrylamide-2-d

Enallene **1f** was prepared following the general procedure GP-1 from the corresponding enyne (234.4 mg, 0.88 mmol). Orange solid (155.9 mg, 67% yield). Two rotamers were observed due to the dynamic rotation of the amide. ^1^H NMR (400 MHz, CDCl_3_) δ 7.80 (t, *J* = 6.4 Hz, 1H RotA), 7.61–7.53 (m, 1H RotA, 1H RotB, 7.52–7.17 (m, 6H RotA, 6H RotB), 6.98–6.91 (m, 1H RotB, 1H RotB non-deuterated 1f), 6.73 (d, *J* = 15.1 Hz, 1H RotA non-deuterated 1f), 6.64–6.55 (m, 1H RotA, 1H RotB), 6.51–6.42 (m, 1H RotA, 1H RotB), 5.39–5.27 (m, 2H RotA, 2H RotB), 4.88–4.79 (m, 2H RotA, 2H RotB). ^13^C NMR (101 MHz, CDCl_3_) δ 202.8, 202.4, 164.9, 151.5, 151.4, 144.3, 137.6, 137.1, 131.1, 131.0, 130.5, 130.4, 128.8, 128.4, 128.0, 127.4, 127.1, 126.2, 114.8, 114.6, 114.3, 114.2, 114.0 (t, *J* = 24.1 Hz), 112.3, 100.2, 99.9, 87.6, 86.8, 49.2, 48.1. ESI-MS calculated for C_17_H_15_DNO_2_ [M + H]^+^; 267.12 found 266.96.

#### 4.1.7. (*E*)-*N*-(4-Fluorobenzyl)-3-(furan-2-yl)-*N*-(propa-1,2-dien-1-yl)acrylamide

Enallene **1g** was prepared following the general procedure GP-1 from the corresponding enyne (311.6 mg, 1.1 mmol). Orange solid (189.6 mg, 61% yield). Two rotamers were observed due to the dynamic rotation of the amide. ^1^H NMR (400 MHz, CDCl_3_) δ 7.77 (t, *J* = 6.4 Hz, 1H RotA), 7.60–7.52 (m, 1H RotA, 1H RotB), 7.45 (d, *J* = 22.8 Hz, 1H RotA, 1H RotB), 7.34–7.18 (m, 2H RotA, 2H RotB), 7.09–6.88 (m, 2H RotA, 4H RotB), 6.70 (d, *J* = 15.0 Hz, 1H RotA), 6.62–6.59 (m, 1H RotA, 1H RotB), 6.49–6.45 (m, 1H RotA, 1H RotB), 5.37–5.33 (m, 2H RotA, 2H RotB), 4.79 (s, 2H RotA, 2H RotB). ^13^C NMR (101 MHz, CDCl_3_) δ 202.7, 202.3, 164.9, 164.8, 163.3 (d, *J* = 8.3 Hz), 160.8 (d, *J* = 8.5 Hz), 151.5, 151.4, 144.4, 133.4, 132.8, 131.3, 130.6, 129.8, 129.7, 127.9, 127.8, 115.8, 115.6, 115.2, 115.1, 115.0, 114.7, 114.1, 114.0, 112.4, 100.1, 99.8, 87.7, 86.9, 48.5, 47.4. ^19^F NMR (565 MHz, CDCl_3_) δ -115.02 (q, *J* = 7.2 Hz, 1F RotA), -115.52 (q, *J* = 7.1 Hz, 1F RotB). ESI-MS calculated for C_17_H_15_FNO_2_ [M + H]^+^; 284.11 found 284.06.

#### 4.1.8. (*E*)-3-(Furan-2-yl)-*N*-(3-methoxybenzyl)-*N*-(propa-1,2-dien-1-yl)acrylamide

Enallene **1h** was prepared following the general procedure GP-1 from the corresponding enyne (295.3 mg, 1.0 mmol). Orange solid (222.6 mg, 75% yield). Two rotamers were observed due to the dynamic rotation of the amide. ^1^H NMR (400 MHz, CDCl_3_) δ 7.79 (t, *J* = 6.4 Hz, 1H RotA), 7.56 (dd, *J* = 15.1, 3.7 Hz, 1H RotA, 1H RotB), 7.48 (s, 1H RotB), 7.41 (s, 1H RotA), 7.30–7.21 (m, 1H RotA, 1H RotB), 6.97–6.69 (m, 4H RotA, 5H RotB), 6.64–6.55 (m, 1H RotA, 1H RotB), 6.51–6.42 (m, 1H RotA, 1H RotB), 5.35 (d, *J* = 9.0 Hz, 2H RotA, 2H RotB), 4.81 (d, *J* = 9.0 Hz, 2H RotA, 2H RotB), 3.80 (s, 3H RotA, 3H RotB). ^13^C NMR (101 MHz, CDCl_3_) δ 202.7, 202.4, 164.9, 160.0, 159.7, 151.5, 144.3, 139.2, 138.8, 131.1, 130.5, 129.9, 129.3, 120.3, 118.4, 114.8, 114.6, 114.3, 113.6, 112.6, 112.5, 112.3, 112.0, 100.3, 99.9, 87.7, 86.8, 55.2, 49.1, 48.1. ESI-MS calculated for C_18_H_18_NO_3_ [M + H]^+^; 296.13 found 296.07.

#### 4.1.9. (*E*)-*N*-Cyclopropyl-3-(furan-2-yl)-*N*-(propa-1,2-dien-1-yl)acrylamide

Enallene **1i** was prepared following the general procedure GP-1 from the corresponding enyne (129.2 mg, 0.6 mmol). Brown oil (86.1 mg, 66% yield). ^1^H NMR (400 MHz, CDCl_3_) δ 7.49–7.46 (m, 3H), 7.18 (d, *J* = 15.3 Hz, 1H), 6.60 (d, *J* = 3.4 Hz, 1H), 6.48 (dd, *J* = 3.4, 1.8 Hz, 1H), 5.36 (d, *J* = 6.5 Hz, 2H), 2.74 (tt, *J* = 7.0, 3.8 Hz, 1H), 1.05–1.00 (m, 2H), 0.88–0.84 (m, 2H). ^13^C NMR (101 MHz, CDCl_3_) δ 203.6, 166.2, 151.7, 144.2, 129.7, 115.8, 114.5, 112.3, 99.6, 85.5, 28.0, 10.1. ESI-MS calculated for C_13_H_14_NO_2_ [M + H]^+^; 216.10 found 216.04.

#### 4.1.10. (*E*)-3-(Benzo[b]thiophen-2-yl)-*N*-benzyl-*N*-(propa-1,2-dien-1-yl)acrylamide

Enallene **1j** was prepared following the general procedure GP-1 from the corresponding enyne (334.4 mg, 1.0 mmol). Brown solid (128.5 mg, 38% yield). Two rotamers were observed due to the dynamic rotation of the amide. ^1^H NMR (400 MHz, CDCl_3_) δ 7.99 (dd, *J* = 15.1, 3.2 Hz, 1H RotA, 1H RotB), 7.84–7.74 (m, 3H RotA, 2H RotB), 7.54–7.23 (m, 8H RotA, 8H RotB), 6.91–6.88 (m, 1H RotA, 1H RotB), 6.68 (d, *J* = 14.9 Hz, 1H RotB), 5.39 (d, *J* = 6.3 Hz, 2H RotA, 2H RotB), 4.85 (d, *J* = 14.5 Hz, 2H RotA, 2H RotB). ^13^C NMR (101 MHz, CDCl_3_) δ 202.7, 164.5, 164.4, 140.1, 139.9, 139.7, 137.6, 137.5, 137.14, 137.06, 128.9, 128.6, 128.4, 128.1, 127.6, 127.2, 126.3, 126.1, 124.9, 124.44, 122.37, 122.4, 118.3, 118.2, 100.1, 87.9, 86.8, 49.3, 48.4. ESI-MS calculated for C_21_H_18_NOS [M + H]^+^; 332.11 found 332.08.

#### 4.1.11. (*E*)-3-(Benzo[b]thiophen-2-yl)-*N*-(3-methylbenzyl)-*N*-(propa-1,2-dien-1-yl)acrylamide

Enallene **1k** was prepared following the general procedure GP-1 from the corresponding enyne (390.37 mg, 1.13 mmol). Yellow solid (226.3 mg, 58% yield). Two rotamers were observed due to the dynamic rotation of the amide. ^1^H NMR (400 MHz, CDCl_3_) δ 8.03–7.95 (m, 1H RotA, 1H RotB), 7.86–7.71 (m, 3H RotA, 2H RotB), 7.50–7.06 (m, 7H RotA, 7H RotB), 6.94–6.86 (m, 2H RotB), 6.68 (d, *J* = 14.9 Hz, 1H RotA), 5.40 (d, *J* = 6.4 Hz, 2H RotA, 2H RotB), 4.86–4.76 (m, 2H RotA, 2H RotB), 2.41–2.33 (m, 3H RotA, 3H RotB). ^13^C NMR (101 MHz, CDCl_3_) δ 202.8, 202.7, 164.5, 164.4, 140.2, 139.9, 139.7, 138.7, 138.1, 137.5, 137.3, 137.1, 137.0, 128.84, 128.81, 128.7, 128.6, 128.51, 128.47, 128.4, 128.3, 128.0, 126.9, 126.1, 125.2, 125.1, 125.0, 124.9, 124.4, 123.4, 122.43, 122.36, 118.4, 118.2, 100.2, 87.8, 86.8, 49.3, 48.4, 21.5, 21.4. ESI-MS calculated for C_22_H_20_NOS [M + H]^+^; 346.13 found 345.88. (*Partial decomposition was observed during the acquisition of NMR spectra*).

#### 4.1.12. (*E*)-3-(Benzofuran-2-yl)-*N*-benzyl-*N*-(propa-1,2-dien-1-yl)acrylamide

Enallene **1l** was prepared following the general procedure GP-1 from the corresponding enyne (414.4 mg, 1.2 mmol). Pale yellow solid (248.8 mg, 60% yield). Two rotamers were observed due to the dynamic rotation of the amide. ^1^H NMR (400 MHz, CDCl_3_) δ 7.81 (t, *J* = 6.4 Hz, 1H RotA), 7.69 (dd, *J* = 15.0, 8.2 Hz, 1H RotA, 1H RotB), 7.60 (dd, *J* = 11.4, 7.7 Hz, 1H RotA, 1H RotB), 7.57–7.19 (m, 8H RotA, 8H RotB), 7.03–6.99 (m, 1H RotA, 2H RotB), 6.94 (d, *J* = 10.2 Hz, 1H RotA, 1H RotB), 5.39–5.34 (m, 2H RotA, 2H RotB), 4.89 (s, 2H RotA, 2H RotB). ^13^C NMR (101 MHz, CDCl_3_) δ 202.9, 202.5, 164.5, 155.4, 153.0, 152.9, 137.5, 137.0, 131.4, 130.7, 128.8, 128.5, 128.4, 128.0, 127.5, 127.2, 126.32, 126.28, 123.3, 121.8, 117.3, 117.2, 111.3, 111.2, 110.9, 100.2, 99.8, 87.7, 86.9, 49.3, 48.3. ESI-MS calculated for C_21_H_17_NNaO_2_ [M + Na]^+^; 338.12 found 338.52.

#### 4.1.13. (*E*)-*N*-Benzyl-3-(3-methylthiophen-2-yl)-*N*-(propa-1,2-dien-1-yl)acrylamide

Enallene **1m** was prepared following the general procedure GP-1 from the corresponding enyne (295.4 mg, 1 mmol). Brown solid (236.3 mg, 80% yield). Two rotamers were observed due to the dynamic rotation of the amide. ^1^H NMR (400 MHz, CDCl_3_) δ 8.03–7.93 (m, 1H RotA, 1H RotB), 7.82 (t, *J* = 6.4 Hz, 1H RotA), 7.41–7.17 (m, 6H RotA, 6H RotB), 6.94–6.83 (m, 1H RotA, 2H RotB), 6.76 (d, *J* = 15.0 Hz, 1H RotB), 6.55 (d, *J* = 14.9 Hz, 1H RotA), 5.37 (d, *J* = 6.3 Hz, 2H RotA, 2H RotB), 4.88–4.77 (m, 2H RotA, 2H RotB), 2.40–2.30 (m, 3H RotA, 3H RotB). ^13^C NMR (101 MHz, CDCl_3_) δ 202.7, 202.6, 165.05, 164.95, 141.2, 137.7, 137.3, 135.5, 135.1, 134.3, 131.3, 128.8, 128.4, 128.0, 127.5, 127.1, 126.4, 126.3, 114.8, 114.5, 100.2, 87.7, 86.8, 49.3, 48.3, 14.2. ESI-MS calculated for C_17_H_18_NOS [M + H]^+^; 296.11 found 295.67.

#### 4.1.14. (*E*)-*N*-Benzyl-*N*-(propa-1,2-dien-1-yl)-3-(thiophen-3-yl)acrylamide

Enallene **1n** was prepared following the general procedure GP-1 from the corresponding enyne (281.4 mg, 1 mmol). Orange oil (202.6 mg, 72% yield). Two rotamers were observed due to the dynamic rotation of the amide. ^1^H NMR (400 MHz, CDCl_3_) δ 7.83–7.77 (m, 2H RotA, 1H RotB), 7.52–7.26 (m, 7H RotA, 8H RotB), 7.16 (d, *J* = 5.2 Hz, 1H RotA), 6.93–6.83 (m, 2H RotB), 6.62 (d, *J* = 15.2 Hz, 1H RotA), 5.35 (d, *J* = 6.4 Hz, 2H RotA, 2H RotB), 4.84 (d, *J* = 12.0 Hz, 2H RotA, 2H RotB). ^13^C NMR (101 MHz, CDCl_3_) δ 202.7, 202.4, 165.3, 165.2, 138.2, 138.1, 137.6, 137.3, 128.9, 128.4, 128.0, 127.9, 127.7, 127.5, 127.1, 126.9, 126.8, 126.1, 125.1, 116.6, 116.5, 100.2, 100.1, 87.7, 86.8, 49.3, 48.2. ESI-MS calculated for C_17_H_16_NOS [M + H]^+^; 282.09 found 281.63.

#### 4.1.15. (*E*)-*N*-Benzyl-3-(1-benzyl-1*H*-pyrrol-2-yl)-*N*-(propa-1,2-dien-1-yl)acrylamide

Enallene **1o** was prepared following the general procedure GP-1 from the corresponding enyne (212.6 mg, 0.6 mmol). Orange oil (98.2 mg, 46% yield). Two rotamers were observed due to the dynamic rotation of the amide. ^1^H NMR (400 MHz, Acetone-*d*6) δ 7.72–7.64 (m, 2H RotA, 2H RotB), 7.34–7.26 (m, 9H RotA, 9H RotB), 7.13–7.06 (m, 3H RotA, 3H RotB), 7.00–6.94 (m, 1H RotA), 6.75–6.67 (m, 1H RotA, 2H RotB), 6.25–6.19 (m, 1H RotA, 1H RotB), 5.40–5.31 (m, 4H RotA, 4H RotB), 4.83–4.77 (m, *J* = 8.6 Hz, 2H RotA, 2H RotB). ^13^C NMR (101 MHz, Acetone-*d*6) δ 202.7, 202.1, 164.5, 138.5, 138.1, 132.0, 129.5, 128.7, 128.5, 128.3, 127.6, 127.4, 127.1, 126.7, 126.6, 126.3, 121.4, 111.9, 111.8, 109.3, 100.4, 99.6, 86.8, 85.9, 50.2, 49.7, 48.3, 47.3. ESI-MS calculated for C_24_H_23_N_2_O [M + H]^+^; 355.18 found 355.10. (*Partial decomposition was observed during the acquisition of NMR spectra*).

#### 4.1.16. (*E*)-*N*-Benzyl-3-(1-methyl-1*H*-indol-2-yl)-*N*-(propa-1,2-dien-1-yl)acrylamide

Enallene **1p** was prepared following the general procedure GP-1 from the corresponding enyne (164.2 mg, 0.5 mmol). Orange solid (79.4 mg, 48% yield). Two rotamers were observed due to the dynamic rotation of the amide. ^1^H NMR (400 MHz, Acetone-*d*6) δ 7.93–7.81 (m, *J* = 19.8, 10.0 Hz, 1H RotA, 2H RotB), 7.58–6.96 (m, 12H RotA, 11H RotB), 5.39 (d, *J* = 6.4 Hz, 2H RotA, 2H RotB), 4.91 (d, *J* = 43.3 Hz, 2H RotA, 2H RotB), 3.89 (s, 3H RotB), 3.80 (s, 3H RotA). ^13^C NMR (101 MHz, Acetone-*d*6) δ 207.9, 207.4, 169.2, 144.3, 143.4, 140.9, 137.2, 136.9, 133.9, 133.8, 133.4, 133.0, 133.0, 132.8, 132.5, 132.4, 131.5, 128.3, 126.1, 125.3, 122.6, 122.4, 115.1, 114.1, 108.2, 105.6, 104.9, 92.3, 91.3, 53.7, 52.7, 34.5. ESI-MS calculated for C_22_H_21_N_2_O [M + H]^+^; 329.17 found 329.60. (*Partial decomposition was observed during the acquisition of NMR spectra*).

### 4.2. General Procedure for the Photocatalytic Reaction Leading to Products ***2** (**GP-2**)*

To a vial charged with substrate **1** (1 equiv., 0.2 mmol) and Ir(p-F-ppy)_3_ (1 mol%), dry DCM (0.1 M) was added through a syringe. The solution was transferred into an NMR tube capped with a rubber septum, and it was placed in an oil bath kept at 40 °C and irradiated with LED stripes for 3 h. Conversion was monitored by TLC, and the mixture was then concentrated in vacuo. The residue was purified by chromatography on silica gel; the catalyst was removed using toluene as an eluent prior to the separation of the desired products (*N*-hexane/EtOAc, under gradient).

#### 4.2.1. 1-Benzyl-1,4,8a,8b-tetrahydro-2*H*-thieno [2,3-g]indol-2-one

Product **2a** was prepared following the general procedure GP-2 from the corresponding enallene (56.7 mg, 0.2 mmol; 392.1 mg, 1.39 mmol). White solid (35.7 mg, 63% yield; 205.9 mg, 52%, 9 h of irradiation). ^1^H NMR (400 MHz, CDCl_3_) δ 7.39–7.19 (m, 5H), 6.33 (dd, *J* = 6.4, 2.6 Hz, 1H), 6.07 (s, 1H), 5.76 (dd, *J* = 6.4, 1.4 Hz, 1H), 5.67 (q, *J* = 3.4 Hz, 1H), 5.15 (d, *J* = 15.8 Hz, 1H), 4.48 (d, *J* = 15.8 Hz, 1H), 3.91 (d, *J* = 10.3 Hz, 1H), 3.63–3.55 (m, 1H), 3.46–3.27 (m, 2H). ^13^C NMR (101 MHz, CDCl_3_) δ 172.3, 157.3, 137.6, 137.5, 128.9, 127.5, 127.4, 127.3, 121.2, 121.1, 115.6, 62.4, 56.1, 45.4, 28.8. ESI-HRMS calculated for C_17_H_16_NOS [M + H]^+^; 282.0948 found 282.0955.

#### 4.2.2. 1-(2-Bromobenzyl)-1,4,8a,8b-tetrahydro-2*H*-thieno [2,3-g]indol-2-one

Product **2b** was prepared following the general procedure GP-2 from the corresponding enallene (71.6 mg, 0.2 mmol). White solid (39.4 mg, 55% yield). ^1^H NMR (400 MHz, CDCl_3_) δ 7.58 (dd, *J* = 7.9, 1.3 Hz, 1H), 7.32–7.24 (m, 1H), 7.19–7.08 (m, 2H), 6.29 (dd, *J* = 6.4, 2.6 Hz, 1H), 6.11 (s, 1H), 5.70 (q, *J* = 3.5 Hz, 1H), 5.58–5.53 (m, 1H), 4.93 (d, *J* = 17.0 Hz, 1H), 4.81 (d, *J* = 17.0 Hz, 1H), 4.02 (d, *J* = 10.3 Hz, 1H), 3.64–3.56 (m, 1H), 3.49–3.33 (m, 2H). ^13^C NMR (101 MHz, CDCl_3_) δ 172.4, 157.7, 137.6, 136.3, 133.0, 128.9, 128.2, 127.9, 127.4, 122.3, 121.1, 115.5, 63.6, 56.0, 45.8, 28.8. ESI-HRMS calculated for C_17_H_15_BrNOS [M + H]^+^; 360.0053 found 360.0051.

#### 4.2.3. 1-(Benzo[d][1,3]dioxol-5-ylmethyl)-1,4,8a,8b-tetrahydro-2*H*-thieno [2,3-g]indol-2-one

Product **2c** was prepared following the general procedure GP-2 from the corresponding enallene (65.7 mg, 0.2 mmol). White solid (33.3 mg, 51% yield). ^1^H NMR (600 MHz, CDCl_3_) δ 6.74 (d, *J* = 7.8 Hz, 1H), 6.69–6.64 (m, 2H), 6.33 (dd, *J* = 6.3, 2.6 Hz, 1H), 6.02 (s, 1H), 5.93 (s, 2H), 5.77 (dd, *J* = 6.4, 1.4 Hz, 1H), 5.65 (q, *J* = 3.9 Hz, 1H), 5.03 (d, *J* = 15.6 Hz, 1H), 4.34 (d, *J* = 15.6 Hz, 1H), 3.88 (d, *J* = 10.3 Hz, 1H), 3.58–3.52 (m, 1H), 3.38 (dt, *J* = 22.1, 4.1 Hz, 1H), 3.30 (dq, *J* = 22.0, 4.1 Hz, 1H). ^13^C NMR (101 MHz, CDCl_3_) δ 172.3, 157.4, 148.2, 147.0, 137.6, 131.3, 127.4, 121.2, 121.1, 120.6, 115.6, 108.4, 107.9, 101.1, 62.3, 56.1, 45.2, 28.7. ESI-HRMS calculated for C_18_H_16_NO_3_S [M + H]^+^; 326.0846 found 326.0851.

#### 4.2.4. 1-Methyl-1,4,8a,8b-tetrahydro-2*H*-thieno [2,3-g]indol-2-one

Product **2d** was prepared following general procedure GP-2 from the corresponding enallene (40.5 mg, 0.2 mmol). White solid (16.9 mg, 42% yield). ^1^H NMR (400 MHz, Acetone-*d*6) δ 6.58 (dd, *J* = 6.4, 2.6 Hz, 1H), 6.11–6.07 (m, 1H), 5.90 (q, *J* = 1.6 Hz, 1H), 5.76–5.72 (m, 1H), 3.94 (d, *J* = 10.3 Hz, 1H), 3.56–3.48 (m, 1H), 3.41–3.36 (m, 2H), 3.08 (s, 3H). ^13^C NMR (101 MHz, Acetone-*d*6) δ 170.7, 156.7, 137.3, 126.8, 121.8, 120.8, 115.9, 64.3, 56.5, 28.14, 28.06. ESI-HRMS calculated for C_11_H_12_NOS [M + H]^+^; 206.0635 found 206.0639.

#### 4.2.5. 1-Benzyl-1,4,8a,8b-tetrahydro-2*H*-furo [2,3-g]indol-2-one

Product **2e** was prepared following general procedure GP-2 from the corresponding enallene (53.0 mg, 0.2 mmol). White solid (27.6 mg, 52% yield). ^1^H NMR (400 MHz, CDCl_3_) δ 7.40–7.20 (m, 5H), 6.53–6.49 (m, 1H), 6.09–6.05 (m, 1H), 5.23–5.18 (m, 2H), 5.03 (d, *J* = 15.6 Hz, 1H), 4.47 (d, *J* = 15.6 Hz, 1H), 3.85 (d, *J* = 9.4 Hz, 1H), 3.44–3.34 (m, 2H), 3.32–3.21 (m, 1H). ^13^C NMR (101 MHz, CDCl_3_) δ 172.1, 157.6, 154.7, 147.4, 137.6, 128.8, 127.53, 127.47, 121.5, 103.3, 94.8, 63.6, 47.8, 45.0, 25.2. ESI-HRMS calculated for C_17_H_16_NO_2_ [M + H]^+^; 266.1176 found 266.1170. (*Partial decomposition was observed during the acquisition of NMR spectra*).

#### 4.2.6. 1-Benzyl-1,4,8a,8b-tetrahydro-2*H*-furo [2,3-g]indol-2-one-3-d

Product **2f** was prepared following the general procedure GP-2 from the corresponding enallene (53.3 mg, 0.2 mmol). White solid (27.2 mg, 51% yield). ^1^H NMR (600 MHz, CDCl_3_) δ 7.34–7.17 (m, 5H), 6.49–6.46 (m, 1H), 6.03 (brs, 1H non-deuterated 2f), 5.19–5.15 (m, 2H), 4.99 (d, *J* = 15.6 Hz, 1H), 4.43 (d, *J* = 15.6 Hz, 1H), 3.81 (d, *J* = 9.8 Hz, 1H), 3.39–3.31 (m, 2H), 3.27–3.19 (m, 1H). ^13^C NMR (101 MHz, CDCl_3_) δ 172.1, 157.6 (non-deuterated 2f), 157.4, 154.7, 147.3, 137.7, 128.8, 127.52, 127.46, 121.5 (non-deuterated 2f), 103.3, 94.8, 63.59 (non-deuterated 2f), 63.56, 47.8, 45.0, 25.22 (non-deuterated 2f), 25.19. ESI-HRMS calculated for C_17_H_15_DNO_2_ [M + H]^+^; 267.1239 found 267.1243. (*Partial decomposition was observed during the acquisition of NMR spectra*).

#### 4.2.7. 1-(4-Fluorobenzyl)-8a,8b-dihydro-1*H*-furo [2,3-g]indol-2(4*H*)-one

Product **2g** was prepared following the general procedure GP-2 from the corresponding enallene (56.7 mg, 0.2 mmol). Pale yellow solid (29.4 mg, 52% yield). ^1^H NMR (400 MHz, CDCl_3_) δ 7.24–7.21 (m, 2H), 7.05–7.00 (m, 2H), 6.53 (dd, *J* = 3.0, 2.1 Hz, 1H), 6.06 (dt, *J* = 2.3, 1.2 Hz, 1H), 5.23–5.19 (m, 2H), 4.98 (d, *J* = 15.5 Hz, 1H), 4.43 (d, *J* = 15.6 Hz, 1H), 3.82 (d, *J* = 9.3 Hz, 1H), 3.42–3.34 (m, 2H), 3.31–3.22 (m, 1H). ^13^C NMR (101 MHz, CDCl_3_) δ 172.1, 163.4, 161.0, 157.7, 154.6, 147.6, 133.5, 133.4, 129.2, 121.5, 115.8, 115.6, 103.2, 94.9, 63.6, 47.7, 44.3, 25.2. ^19^F NMR (565 MHz, CDCl_3_) δ -114.78 (td, *J* = 8.7, 4.4 Hz). ESI-HRMS calculated for C_17_H_15_FNO_2_ [M + H]^+^; 284.1082 found 284.1081.

#### 4.2.8. 1-(3-Methoxybenzyl)-8a,8b-dihydro-1*H*-furo [2,3-g]indol-2(4*H*)-one

Product **2h** was prepared following the general procedure GP-2 from the corresponding enallene (59.1 mg, 0.2 mmol). White solid (29.3 mg, 50% yield). ^1^H NMR (400 MHz, CDCl_3_) δ 7.25 (t, *J* = 7.8 Hz, 1H), 6.84–6.78 (m, 3H), 6.52 (dd, *J* = 3.0, 2.1 Hz, 1H), 6.06 (dt, *J* = 2.3, 1.2 Hz, 1H), 5.23–5.20 (m, 2H), 4.99 (d, *J* = 15.6 Hz, 1H), 4.44 (d, *J* = 15.6 Hz, 1H), 3.86 (dd, *J* = 9.3, 1.4 Hz, 1H), 3.80 (s, 3H), 3.43–3.35 (m, 2H), 3.30–3.21 (m, 1H). ^13^C NMR (101 MHz, CDCl_3_) δ 172.0, 160.0, 157.6, 154.7, 147.4, 139.3, 129.8, 121.5, 119.7, 113.1, 112.8, 103.4, 94.8, 63.6, 55.3, 47.8, 44.9, 25.2. ESI-HRMS calculated for C_18_H_18_NO_3_ [M + H]^+^; 296.1282 found 296.1288.

#### 4.2.9. 1-Cyclopropyl-8a,8b-dihydro-1*H*-furo [2,3-g]indol-2(4*H*)-one

Product **2i** was prepared following the general procedure GP-2 from the corresponding enallene (43.1 mg, 0.2 mmol). Ocher solid (19.3 mg, 44% yield). ^1^H NMR (400 MHz, Acetone-*d*6) δ 6.76 (dd, *J* = 3.0, 2.3 Hz, 1H), 5.80 (dt, *J* = 2.4, 1.3 Hz, 1H), 5.77 (ddd, *J* = 3.2, 2.2, 1.1 Hz, 1H), 5.20 (dddd, *J* = 4.8, 3.8, 2.8, 1.2 Hz, 1H), 3.90 (dd, *J* = 9.9, 1.3 Hz, 1H), 3.47–3.39 (m, 1H), 3.37–3.22 (m, 2H), 2.65–2.59 (m, 1H), 0.94–0.87 (m, 1H), 0.82–0.74 (m, 3H). ^13^C NMR (101 MHz, Acetone-*d*6) δ 170.9, 157.0, 155.1, 146.8, 121.5, 104.6, 94.8, 64.3, 48.0, 24.7, 23.0, 7.7, 4.6. ESI-HRMS calculated for C_13_H_14_NO_2_ [M + H]^+^; 216.1020 found 216.1013.

#### 4.2.10. 1-Benzyl-10b,10c-dihydro-1*H*-benzo [4,5]thieno [2,3-g]indol-2(4*H*)-one

Product **2j** was prepared following the general procedure GP-2 from the corresponding enallene (66.3 mg, 0.2 mmol). Pale brown solid (42.1 mg, 63% yield). ^1^H NMR (400 MHz, Acetone-*d*6) δ 7.51 (dd, *J* = 7.7, 1.0 Hz, 1H), 7.31–7.23 (m, 5H), 7.16–7.08 (m, 3H), 6.11 (dt, *J* = 2.2, 1.1 Hz, 1H), 5.83 (dt, *J* = 5.0, 3.1 Hz, 1H), 5.31 (d, *J* = 15.9 Hz, 1H), 4.59 (d, *J* = 15.9 Hz, 1H), 4.23–4.18 (m, 2H), 3.56–3.49 (m, 1H), 3.44–3.35 (m, 1H). ^13^C NMR (101 MHz, Acetone-*d*6) δ 173.9, 160.2, 139.9, 138.0, 136.9, 136.8, 128.7, 128.6, 127.3, 127.1, 126.4, 124.6, 121.9, 120.8, 115.8, 63.5, 56.0, 47.0. ESI-HRMS calculated for C_21_H_18_NOS [M + H]^+^; 332.1104 found 332.1102.

#### 4.2.11. 1-(3-Methylbenzyl)-1,4,10b,10c-tetrahydro-2*H*-benzo [4,5]thieno [2,3-g]indol-2-one

Product **2k** was prepared following the general procedure GP-2 from the corresponding enallene (45.6 mg, 0.13 mmol). White solid (20.3 mg, 45% yield). ^1^H NMR (400 MHz, CDCl_3_) δ 7.45 (d, *J* = 7.7 Hz, 1H), 7.30–7.20 (m, 2H), 7.18–7.10 (m, 2H), 7.04 (d, *J* = 7.6 Hz, 1H), 6.86–6.79 (m, 2H), 6.13 (s, 1H), 5.76–5.73 (m, 1H), 5.40 (d, *J* = 15.5 Hz, 1H), 4.47 (d, *J* = 15.6 Hz, 1H), 4.16–4.11 (m, 2H), 3.54–3.45 (m, 1H), 3.39–3.29 (m, 1H), 2.28 (s, 3H). ^13^C NMR (101 MHz, CDCl_3_) δ 174.9, 160.2, 140.0, 138.5, 137.6, 137.1, 136.3, 128.8, 128.6, 128.3, 128.2, 126.5, 124.6, 122.1, 121.3, 114.8, 63.7, 56.0, 47.4, 28.9, 21.4. ESI-HRMS calculated for C_22_H_20_NOS [M + H]^+^; 346.1261 found 346.1266.

#### 4.2.12. 1-Benzyl-10b,10c-dihydro-1*H*-benzofuro [2,3-g]indol-2(4*H*)-one

Product **2l** was prepared following the general procedure GP-2 from the corresponding enallene (63.1 mg, 0.2 mmol). Pale orange solid (28.0 mg, 45% yield). ^1^H NMR (400 MHz, Acetone-*d*6) δ 7.52 (d, *J* = 7.4 Hz, 1H), 7.33–7.21 (m, 6H), 7.00 (td, *J* = 7.6, 1.1 Hz, 1H), 6.95 (dd, *J* = 8.1, 1.0 Hz, 1H), 6.13 (dt, *J* = 2.4, 1.2 Hz, 1H), 5.38 (dt, *J* = 5.6, 2.9 Hz, 1H), 5.26 (d, *J* = 16.1 Hz, 1H), 4.76 (d, *J* = 16.1 Hz, 1H), 4.19 (dd, *J* = 9.9, 1.2 Hz, 1H), 4.13–4.09 (m, 1H), 3.50 (dddd, *J* = 21.3, 4.4, 2.0, 0.9 Hz, 1H), 3.39–3.31 (m, 1H). ^13^C NMR (101 MHz, Acetone-*d*6) δ 172.8, 159.2, 158.6, 154.3, 138.2, 129.3, 128.6, 127.2, 127.1, 126.5, 125.6, 122.1, 121.5, 109.8, 95.3, 63.2, 48.4, 45.9, 24.8. ESI-HRMS calculated for C_21_H_18_NO_2_ [M + H]^+^; 316.1333 found 316.1337.

#### 4.2.13. 1-Benzyl-8a-methyl-8a,8b-dihydro-1*H*-thieno [2,3-g]indol-2(4*H*)-one

Product **2m** was prepared following the general procedure GP-2 from the corresponding enallene (59.1 mg, 0.2 mmol). Pale yellow solid (42.2 mg, 71% yield). ^1^H NMR (400 MHz, CDCl_3_) δ 7.35–7.21 (m, 5H), 6.27 (d, *J* = 6.3 Hz, 1H), 6.10 (dt, *J* = 2.5, 1.3 Hz, 1H), 5.85 (dd, *J* = 6.4, 1.0 Hz, 1H), 5.64–5.62 (m, 1H), 5.29 (d, *J* = 15.6 Hz, 1H), 4.21 (d, *J* = 15.6 Hz, 1H), 4.07 (s, 1H), 3.42–3.32 (m, 1H), 3.33–3.25 (m 1H), 0.93 (s, 3H). ^13^C NMR (101 MHz, CDCl_3_) δ 173.0, 156.3, 143.3, 137.2, 128.8, 127.7, 127.6, 127.4, 124.6, 121.8, 115.2, 65.0, 55.7, 45.2, 28.1, 18.4. ESI-HRMS calculated for C_18_H_18_NOS [M + H]^+^; 296.1104 found 296.1101.

#### 4.2.14. 1,6-Dibenzyl-4,5,6,8-tetrahydropyrrolo [3,4-f]indol-7(1*H*)-one

Product **2o** was prepared following the general procedure GP-2 from the corresponding enallene (70.9 mg, 0.2 mmol). White solid (40.4 mg, 57% yield). ^1^H NMR (400 MHz, Acetone-*d*6) δ 7.36–7.25 (m, 8H), 7.13–7.10 (m, 2H), 6.84 (d, *J* = 2.8 Hz, 1H), 5.99 (d, *J* = 2.8 Hz, 1H), 5.18 (s, 2H), 4.62 (s, 2H), 3.88 (t, *J* = 2.2 Hz, 2H), 3.47 (t, *J* = 6.8 Hz, 2H), 3.27–3.20 (m, 2H). ^13^C NMR (101 MHz, Acetone-*d*6) δ 170.3, 149.6, 139.1, 138.5, 128.6, 128.4, 127.8, 127.1, 126.6, 124.4, 121.4, 113.9, 105.7, 52.0, 49.9, 45.3, 24.5, 19.6. ESI-HRMS calculated for C_24_H_23_N_2_O [M + H]^+^; 355.1805 found 355.1804.

#### 4.2.15. 2-Benzyl-5-methyl-1,2,5,10-tetrahydropyrrolo [3,4-b]carbazol-3(4*H*)-one

Product **2p** was prepared following the general procedure GP-2 from the corresponding enallene (49.3 mg, 0.15 mmol). White solid (36.3 mg, 74% yield). ^1^H NMR (400 MHz, CDCl_3_) δ 7.49 (d, *J* = 7.8 Hz, 1H), 7.40–7.22 (m, 7H), 7.15–7.11 (m, 1H), 4.74 (s, 2H), 3.92 (d, *J* = 8.4 Hz, 2H), 3.74 (s, 3H), 3.67 (s, 4H). ^13^C NMR (101 MHz, CDCl_3_) δ 171.1, 149.1, 137.5, 132.8, 128.84, 128.80, 128.1, 127.6, 126.1, 121.3, 119.1, 117.8, 108.9, 105.0, 52.6, 46.2, 29.5, 23.4, 20.4. ESI-HRMS calculated for C_22_H_21_N_2_O [M + H]^+^; 329.1649 found 329.1646.

## 5. Conclusions

We reported a simple method for the preparation of fused tricyclic products from enallene derivatives bearing a heteroaryl substituent on the alkene partner. The sequence involves sequential, site-selective [2+2] photocycloaddition between the olefin and the terminal double bond of the cumulated partner. Steric-strain-driven retro- [2+2] ring opening paves the way for the dearomatization of the heteroaryl ring, eventually affording the desired product with high diastereocontrol. This method combines a good functional group tolerance with the presence of valuable functionalities in the final compound, such as skipped 1,4,7-triene units that may represent relevant synthetic handles for further functionalization. We thus anticipate the use of the present strategy in the future and that present results will further increase interest in the development of mild dearomatization protocols.

## Data Availability

All data supporting reported results can be found in the Appendix A. The accession number for the X-ray data (cif files) reported in this paper is CCDC: 2312634, 2320886.

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
