# Peer review of "Visible-Light-Promoted Tandem Skeletal Rearrangement/Dearomatization of Heteroaryl Enallenes"

_molecules, 2024, doi:10.3390/molecules29030595_

Round 1

Reviewer 1 Report

Comments and Suggestions for Authors

The authors describe a tandem skeletal rearrangement promoted by  visible light, which leads to a complex three-dimensional arhitecture. The present work is an extension of earlier research and represents a nice addition. The purity of the starting material is not ideal and could in principle be improved (according to 1H and 13C spectra). In general, the study is well conducted and structured. The conclusions are well-founded.

Comments on the Quality of English Language

Minor editing of English language required.

Author Response

Please find attached a point-by-point list of answers and modifications made to address all of the referee's comments

Reviewer 2 Report

Comments and Suggestions for Authors

This paper describes a facile method for the preparation of fused tricyclic products from enallene derivatives bearing a heteroaryl substituent on the alkene partner. It is an interesting process. I suggest it to be published in Molecules after addressing the following questions

 1)  For the substrate scopes in Figure 2, the R groups on nitrogen atom in substrate 1 were examined just by benzyl or related groups. The other electron-withdrawing groups such as Ts-, Boc- or acyl groups are suggested to be examined, and added in the text.

2) From the results of N-heterocyles in Figure 2, no dearomatization on the pyrrole ring occurred and the α,β-C=C bond was arranged in different way after the reaction comparing with the results of O- & S-heterocycles. Thus, different reaction mechanism definitely occurred. I suggested the author added the mechanisn in the paper.

Author Response

(The authors gave the same response as above.)

Reviewer 3 Report

Comments and Suggestions for Authors

The paper reports a simple photochemical procedure for the synthesis of complex  fused  tricycles. The authors used an Iridium- based photosensitizer as a photocatalyst, and about 20 enallene compounds were studied. It is assumed that the reaction begins with a triplet energy transfer from the photocatalyst to the substrate. The fluorinated Iridium complex having the highest triplet energy provided the best result. Interestingly, replacement of the benzyl group (at the nitrogen atom) with a methyl group resulted in a lower yield (2d), while introduction of a methyl group into the thienyl ring gave the highest yield (2m). Thienyl and furyl derivatives showed similar results, but the pyrrole derivative gave a different tricycle. It is worth noting the drastic effect of temperature (table in the Supporting Information, entries 11, 23, 24).  Given this fact, "or at low temperature (10 °C)" can be added to the authors' statement "no reaction occurred without either the sensitizer or light (line 108)".

According to DFT calculations, the cascade reaction starts in the triplet state with the formation of the triplet biradical intermediate II, which produces the ground state intermediate III; formally, this is a stepwise [2+2] cycloaddition. Intermediate III then undergoes a C-C bond cleavage giving singlet biradical intermediate IV. Formally, this is the first step of retro-cycloaddition with olefin metathesis, which follows a different pathway than direct [2+2] cycloaddition. (It can be assumed that in the pyrrole derivative the first stage of retro-cycloaddition follows the same pathway as the direct [2+2] cycloaddition without metathesis of olefins, so another tricycle is finally obtained.) It is likely that C-C bond cleavage is the rate-limiting process of the cascade; if so, this could explain the effect of temperature (see above). The last step of the cascade is the dearomative cyclization of biradical IV to form tricycle 2. This step has a lower barrier (10.9 kcal/mol) compared to C-C bond cleavage (16.2 kcal/mol).

The authors also discuss the possibility of activating the enamine group in intermediate III by triplet energy transfer from the photocatalyst, but I agree with the authors that this is a less feasible alternative; this requires stepwise two-photon absorption.

Author Response

(The authors gave the same response as above.)

Reviewer 4 Report

Comments and Suggestions for Authors

In this paper, the authors present an intriguing method for coupling allene and alkene, resulting in a cyclized product with the potential for carbon rearrangement. The conceptual framework exhibited in this work holds significance across various areas of methodology development. To enhance the overall impact of this study, I propose a brief discussion on the importance of fully saturated carbon in influencing the success rate of drug trials. References such as F(sp3) [Drug Discov. Today 25, 1839–1845 (2020)] and recent synthetic efforts [Nat. Chem. 12, 310–317 (2020).] [Nat Catal (2024). https://doi.org/10.1038/s41929-023-01073-5] could be incorporated to draw parallels. Additionally, comparing this work with other methods in Scheme 1, including relevant parameters like the scope number and scale, would provide valuable insights.

For the experimental setup, detailing the lamp specifications and the distance to the test tube is crucial. Since the reaction takes place in a silicone oil bath, information about oil absorption should be included. While the lamp's wavelength is mentioned as 455 nm, providing a full transmission spectrum especially the width of the peak for better reproducibility by other researchers is recommended. The absence of a procedure section for the target reaction is a notable gap.

Considering factors like dilution and types of photocatalysts (PCs), the inclusion of absorption spectra for both the substrate and PCs would enhance the discussion. Further exploration of the influence of parameters like the lamp's maxima on the reaction, along with a detailed discussion on the impact of dilution and the types of PCs used, is advisable.

The proposed mechanism primarily relies on DFT calculations, necessitating experimental validation to distinguish between energy transfer and electron transfer mechanisms between the PC and the substrate. The author suggest that the energy transfer firstly go to the allene moiety, while addition compounds containing allene and the alkene should be prepare and investigate their quenching effects on the PC's emission. Integrating this experimental support would strengthen the credibility of the proposed mechanism.

Author Response

(The authors gave the same response as above.)

Round 2

Reviewer 4 Report

Comments and Suggestions for Authors

Thanks for the authors' detailed explanation on the issues and perform addtional mesurement to improve the quality of the paper. I suggest to accept the paper as present.